# MARC: Memory-Augmented RL Token Compression for Efficient Video Understanding

**Peiran Wu**[1,2]*[†] **Zhuorui Yu**[2][†] **Yunze Liu**[2][‡] **Chi-Hao Wu**[2]**, Enmin Zhou**[2]**, Junxiao Shen**[1,2]
[1]University of Bristol    [2]Memories.ai Research

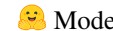 Project Web   Code   Model

## Abstract

The rapid progress of large language models (LLMs) has laid the foundation for multimodal models. Nevertheless, visual language models (VLMs) still face significant computational overhead when scaled from images to the video domain. When video data is too large (due to high frame rates and long durations), the inference cost of models increases sharply. This severely hinders their deployment and application in environments that require rapid responses and have limited computation resources. Token compression for input videos is one of the promising directions, as effective compression schemes can significantly reduce computational overhead. Most existing compression methods are based on training-free token merging strategies in either the spatial or temporal dimension. Although these methods reduce computational overhead, their training-free nature inevitably leads to information loss during token compression, resulting in a significant performance drop. To address these challenges, we propose a Memory-Augmented Reinforcement Learning-based Token Compression (MARC) method for efficient video understanding that integrates structured retrieval with RL-based distillation. Our proposed MARC is a retrieve then compress method, which employs a Visual Memory Retriever (VMR) tool and a Compression Group Relative Policy Optimization (C-GRPO) training strategy. The Visual Memory Retriever first segments videos into event-level fragments and selects query-relevant clips. The C-GRPO distills reasoning ability from a Teacher Network to a Student Network by encouraging the output of the student network to match the performance of the teacher network. Extensive experiments on six video benchmarks demonstrate that our compression method achieves nearly identical accuracy to the 64-frame Qwen2.5-VL-3B baseline while using only one frame's worth of tokens as input, resulting in a 95% reduction in visual tokens. Moreover, our approach reduces GPU memory usage by 72% and generation latency by 23.9%. These results demonstrate the strong potential of our compression method as a robust solution for RL-based post-training compression of large-scale models, enabling practical deployment in latency-sensitive and resource-constrained applications such as real-time video question answering, surveillance, and autonomous driving.

## 1 Introduction

Recent advances in large language models (LLMs) have enabled visual language models (VLMs) to reason over multimodal inputs that combine text and images (Liu et al., 2023; Zhu et al., 2025; Bai et al., 2025). Although early applications primarily targeted short context image tasks, the demand for long context video understanding has dramatically increased computational costs. A single image may necessitate thousands of tokens; this computational load is further magnified when extending the model to high frame rate, long-duration video content. This overhead introduces significant latency and memory bottlenecks, hindering the deployment of VLMs in latency-sensitive and resource-constrained applications such as autonomous driving and surveillance systems. To

---

*Project Leader. Work done during an internship at Memories.ai Research.
[†]Equal contribution.
[‡]Corresponding author.

mitigate these challenges, token compression techniques have been explored, with training-free visual token compression as one of the most effective strategies (Li et al., 2025a; Yang et al., 2025b; Zhang et al., 2024). Despite the fact that these methods reduce computational overhead, their training-free approach inherently causes a considerable amount of information loss during token compression, leading to a significant drop in performance.

To overcome these challenges, we propose **MARC**, a Memory-Augmented RL Token Compression method for efficient video understanding. Our approach is a **retrieve and then compress** method, which first uses a **visual memory retriever (VMR)** to identify the most relevant event segments from a video. Following this, we introduce a novel deep compression technique, **Compression Group Relative Policy Optimization (C-GRPO)**, to further compress these retrieved visual memories. This enables us to reduce each video to a token count equivalent to that of a single image while maintaining performance comparable to the uncompressed video.

Specifically, the design of our **Visual Memory Retriever (VMR)** was inspired by insights from cognitive science and neuroimaging. Most existing methods handle temporal and spatial redundancy independently (Wang et al., 2024a; Liu et al., 2025; Song et al., 2024), overlooking the temporally organized and context-aware characteristics of human visual memory. Cognitive science suggests that humans segment continuous experiences into discrete events and recall them through episodic memory, reinstating both low and high-level perceptual features (James et al., 1890; Hebb, 1968; Damasio, 1989; McClelland et al., 1995). Neuroimaging studies further show reengagement of visual cortical regions during memory retrieval (Favila et al., 2022), while event segmentation theory emphasises contextual shifts as natural anchors for recall (Li et al., 2025b). Motivated by these principles, we propose a Visual Memory Retriever that partitions videos into semantically coherent event-level segments and retrieves query-relevant fragments. These fragments are rearranged and sampled, functioning as structured "episodic memories" for downstream reasoning and enabling more human-like temporal processing. By adopting this retrieve-then-compress approach, it dramatically reduces the computational burden and mitigates the negative effects of redundant information on compression quality.

To reduce a video's token count to that of a single frame while ensuring maximum performance after compression, we proposed a post-training compression algorithm based on reinforcement learning, **Compression Group Relative Policy Optimization (C-GRPO)**, which is applied after finding the most relevant memory fragments. The traditional GRPO (Shao et al., 2024) algorithm is used to enhance the model's reasoning capabilities. We have customized and improved its training framework, reward design, and training strategy, and for the first time, propose C-GRPO. This allows the Student Network to retain Teacher-level reasoning ability under aggressive compression, ensuring robustness while drastically lowering computational cost. Specifically, our C-GRPO transfers reasoning ability from a Teacher Network with 64 frames as input, to the Student Network with just one frame's worth of tokens as input. By integrating structured retrieval with RL-based compression, our framework bridges efficiency and accuracy, providing both cognitive grounding and practical scalability.

We conduct extensive experiments across six benchmarks covering both video reasoning and general video understanding. Our framework achieves nearly identical mean performance to the 64-frame Qwen2.5-VL-3B baseline (42.20 vs. 42.21) while using only a single frame, corresponding to just 4.71% of the original visual tokens. Ablation studies further validate the role of each component: Visual Memory Retriever alone boosts baseline accuracy, while C-GRPO ensures stable performance retention under extreme compression. Combined, they yield superior results, with substantial gains on challenging benchmarks such as TempCompass and MVBench. Moreover, efficiency evaluations demonstrate a 72% reduction in GPU memory usage and 23.9% lower generation latency, enabling deployment in real-world scenarios with strict resource constraints.

In summary, our contributions are threefold:

- We propose **MARC**, a novel framework for efficient video understanding. By deftly integrating a structured visual retrieval mechanism with a powerful reinforcement learning based token compression algorithm, our approach achieves exceptional efficiency while preserving high performance.

- We propose **Compression Group Relative Policy Optimization (C-GRPO)**, the first post-training reinforcement learning (RL) strategy specifically designed for video token

compression. C-GRPO transfers the complex reasoning ability from a high token "Teacher" network to a low token "Student" network.

- Our extensive experiments across six benchmarks demonstrate that MARC achieves both superior performance and exceptional efficiency. By reducing **GPU memory usage by 72%** and cutting **generation latency by 23.9%**, our approach maintains performance while enabling deployment in resource-constrained applications.

## 2 RELATED WORK

**Video Compression for Large Language Models.** Recent advances in multimodal large language models (MLLMs) have greatly expanded their applicability to a wide range of video understanding tasks (Bai et al., 2025; Li et al., 2024a). These models generally process videos by employing powerful pre-trained visual encoders such as CLIP (Radford et al., 2021) and SigLIP (Zhai et al., 2023) to transform sampled video frames into visual tokens that can be fed into the language model. This design allows MLLMs to integrate visual and textual information effectively, enabling tasks such as video captioning, temporal reasoning, and question answering. However, when dealing with long or high-resolution videos, practical limitations such as restricted context length, GPU memory constraints, and increased computational cost create a challenging trade-off between the number of tokens per frame and the total number of frames processed. To address these challenges, prior approaches have explored compression techniques (Song et al., 2024), adaptive pruning mechanisms (Wang et al., 2024a), and frame selection strategies during inference (Wang et al., 2024b). While these methods can alleviate computational overhead, they often suffer from substantial performance degradation, particularly when critical temporal or spatial information is discarded. In contrast, our work proposes a novel reinforcement learning based distillation framework that substantially reduces the required number of visual tokens without sacrificing accuracy. By aligning compressed representations with the reasoning ability of a 1fps sampling teacher model, our approach results in faster inference, lower GPU memory usage, and improved efficiency for real world video understanding applications.

**Video Retrieval Augmented Generation.** Video-RAG is a specialized branch of MM-RAG, with its core function being the utilization of video corpora for knowledge retrieval and subsequent generation (Lewis et al., 2020; Jeong et al., 2025). Based on the primary methods for integrating videos with vLLMs, we can categorize existing Video-RAG architectures into the following: **Auxiliary Text Enhancement:** This category of methods aims to circumvent the challenges of directly processing dense video frames by converting video content into concise, query-auxiliary text (Wang et al., 2022; Edge et al., 2024; Pan et al., 2023). This auxiliary text can be spoken content generated by Automatic Speech Recognition (ASR), on-screen text extracted via Optical Character Recognition (OCR), or visual descriptions produced by object detection. This "text-based" concept greatly simplifies the ingestion process and significantly reduces computational overhead, thereby making long video comprehension possible. **Corpus Retrieval:** This paradigm focuses on dynamically retrieving video clips or entire videos from a large video corpus that are relevant to a given query. These retrieved contents are then fed to a generator as a knowledge source (Luo et al., 2021; Ren et al., 2025). This method is particularly suited for queries that require finding specific events or information from a massive video library. **Agent-Based Systems:** These frameworks, exemplified by VideoAgent (Wang et al., 2024b) and M3-Agent (Long et al., 2025), use a large language model (LLM) as a core agent to mimic a human's multi-round reasoning process. The LLM agent iteratively plans, retrieves, and refines information from the video, using tools such as Visual Language Models (VLMs) and Contrastive Language Image Pretraining (CLIP) to assist in decision-making until the question is fully answered. This approach is especially effective for long video question answering that requires complex, multi-step reasoning. In this paper, we adopt the video corpus retrieval scheme. By using efficient video segmentation and retrieval, we can effectively and significantly reduce the number of input tokens and minimize unnecessary computational overhead.

## 3 MEMORY-AUGMENTED RL DISTILLATION

### 3.1 VISUAL MEMORY RETRIEVER

The core principle of the Visual Memory Retriever is to prioritize the retrieval of the most task-relevant video segments before subsequent compression. This **retrieve then compress** strategy

significantly reduces the computational burden while effectively eliminating the negative impact of redundant information on compression quality. This retriever is engineered to transform a continuous video stream into a structured, searchable memory bank, enabling efficient retrieval of relevant visual information to support complex downstream tasks such as video based question answering.

### 3.1.1 EVENT-BASED VIDEO SEGMENTATION

Unlike conventional methods that rely on fixed length temporal windows, our approach employs an event based video segmentation module (Soucek & Lokoc, 2024) to partition long videos into semantically coherent short clips. This module leverages a deep event detection network that analyzes the video stream to identify significant temporal boundaries, such as scene changes, topic shifts, or the commencement of new actions. Each resulting clip, or visual memory fragment, encapsulates a complete and meaningful event, thereby preserving the contextual integrity of the original video. This event-centric approach dramatically reduces the search space for subsequent retrieval steps and ensures that each retrieved fragment is a complete and self contained unit of information.

### 3.1.2 MEMORY RETRIEVAL

The next stage is the retrieval of memory. We map both the inferred query representation and the visual memory fragments into a shared, high-dimensional latent space using an embedding model (Bolya et al., 2025). This space is learned using a contrastive learning framework, ensuring that semantically similar query fragment pairs are located in close proximity. The search process is performed across the entire corpus of visual memory fragments that are semantically related to the query. This step utilizes a highly optimized nearest neighbor search algorithm on the pre-indexed fragment embeddings, allowing for efficient filtering. The final output is an ordered list of the top-$k$ visual memory fragments, which are then passed to a downstream compression model. This retriever provides the LLM with the precise visual evidence required to ground its response, thereby mitigating hallucination and enabling true video-based reasoning.

## 3.2 RL-BASED VIDEO TOKEN COMPRESSION

Building on the Visual Memory Retriever (VMR) in Section 3.1, which transforms long videos into a small set of query-relevant, event-level segments, we first introduce a memory-aware temporal compression layer that is tailored to these retrieved segments. Then, we propose the Compression Group Relative Policy Optimization to maintain performance despite extreme token compression.

### 3.2.1 MEMORY-AWARE TEMPORAL COMPRESSION LAYER

Rather than treating compression as a generic, training-free pre-processing step, our design exploits the structure imposed by VMR: we first preserve short-range temporal coherence inside each retrieved segment, then perform cross-segment consolidation. This memory-first strategy ensures that compression removes redundancy where it is most prevalent (nearby frames within the same episode) while keeping the event evidence that VMR deemed relevant for downstream reasoning. Concretely, we extend cosine similarity based frame merging (in the spirit of prior temporal ToMe methods such as MovieChat (Song et al., 2024)) into a two stage, memory aware procedure that (i) merges highly similar consecutive frames inside each short-term segment to retain local dynamics and (ii) applies a global, light weight consolidation only when the token budget still exceeds the target. This coupling to VMR is key: the compressor is not a standalone heuristic but an intent-aligned module that respects the event boundaries and ranking produced by VMR, thereby preserving the most causally useful frames for QA.

As shown in Figure 1, we first obtain $k$ top-ranked segments from VMR and uniformly sample them at 1 fps to obtain $N$ frames. Each frame is encoded by a visual encoder (e.g., ViT) into patch-level hidden states. Let

$$\mathcal{H} = \{\mathbf{h}_1, \mathbf{h}_2, \ldots, \mathbf{h}_T\}, \quad \mathbf{h}_i \in \mathbb{R}^d, \tag{1}$$

denote the sequence of $T$ visual tokens, where $T = N \cdot P$ and $P$ is the number of patches per frame. We then partition $\mathcal{H}$ along the temporal axis into short-term memory windows of length $m$ frames (intuitively, contiguous frames within one episode); the $j$-th window contains frames indexed from

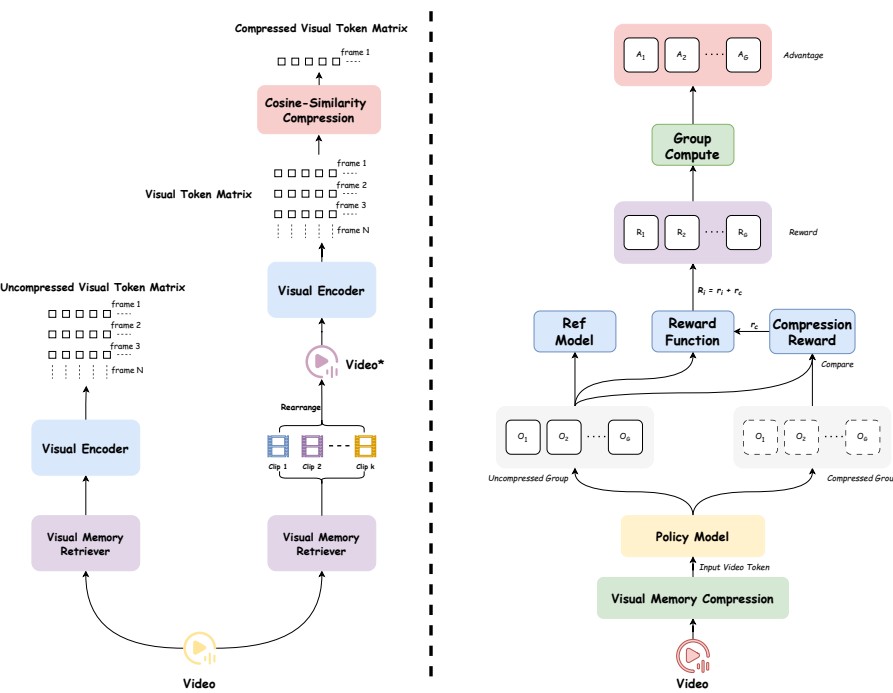

Figure 1: **Left:** Visual Memory Compression. We processed the original video using two approaches: (1) Employing a visual memory retriever to search the video, subsequently reconstructing a new video and compressing its visual features; (2) Also employing a visual memory retriever to search the video, then sampling the original video before feeding it into a visual encoder to obtain uncompressed visual features. **Right:** Compression Group Relative Policy Optimization (C-GRPO) Distillation Framework. Here, **O** denotes the outputs from different groups, **R** the corresponding rewards, and **A** the normalized advantages. A compression reward $r_c$ is introduced to encourage compressed inputs to retain the reasoning ability of the uncompressed teacher model.

$(j-1)m + 1$ to $jm$ (inclusive):

$$\mathcal{S}_j = \{\mathbf{H}_{(j-1)m+1}, \ldots, \mathbf{H}_{jm}\}, \quad j = 1, \ldots, \left\lceil \tfrac{N}{m} \right\rceil, \tag{2}$$

where each $\mathbf{H}_t \in \mathbb{R}^{P \times d}$ stacks the $P$ patch tokens of frame $t$. Inside each $\mathcal{S}_j$ we iteratively merge the two most similar consecutive frame embeddings $\mathbf{H}_a$ and $\mathbf{H}_b$ (consecutive in time), where similarity averages patch-aligned cosine scores (the ViT grid provides a natural patch correspondence):

$$\text{sim}(\mathbf{H}_a, \mathbf{H}_b) = \frac{1}{P} \sum_{p=1}^{P} \frac{\mathbf{h}_a^{(p)} \cdot \mathbf{h}_b^{(p)}}{\|\mathbf{h}_a^{(p)}\| \|\mathbf{h}_b^{(p)}\|}. \tag{3}$$

The two frames are replaced by their mean representation,

$$\mathbf{H}_{\text{merge}} = \tfrac{1}{2} (\mathbf{H}_a + \mathbf{H}_b), \tag{4}$$

and this process repeats until the retained frames in $\mathcal{S}_j$ reach the budget

$$n_j = \max(1, \lfloor (1 - \rho) \cdot |\mathcal{S}_j| \rfloor), \tag{5}$$

where $\rho \in (0, 1)$ is the overall compression ratio (smaller $\rho$ keeps more frames). Finally, we concatenate all compressed segments into $\mathcal{H}' = \{\mathbf{H}'_1, \ldots, \mathbf{H}'_{N'}\}$; if $N' > N_{\text{target}} = \lfloor (1 - \rho)N \rfloor$, we perform a light cross segment merge (same averaging rule) so that local episode structure is preserved first and global pruning acts only as a last resort. The resulting $\mathcal{H}'$ is thus a temporally compressed, memory aware token sequence (with updated grid $\text{THW}'$) that is well aligned with the VMR selected evidence and ready for subsequent transformer layers.

### 3.2.2 COMPRESSION GROUP RELATIVE POLICY OPTIMIZATION (C-GRPO)

We formulate the compression process as a distillation problem: a full-frame teacher provides the reference behaviour, while a single-frame student learns to match its reasoning quality under an aggressively reduced token budget. Standard GRPO (Shao et al., 2024) enforces answer correctness and format but does not explicitly couple student performance to its full-frame counterpart; in contrast, our **C-GRPO** adds a retention alignment reward that directly encourages compressed inputs to preserve teacher-level performance, as illustrated in Figure 1. Formally, let $a_{\text{full}}$ be the average reward with 64 frame inputs and $a_{\text{comp}}$ the reward with the compressed input; the retention ratio

$$\eta = \frac{a_{\text{comp}}}{a_{\text{full}}}, \tag{6}$$

quantifies how much of the teacher's performance the student retains under compression. We then shape the objective with a compression reward

$$r_c = \alpha \cdot \max(0, \ \eta - \tau), \tag{7}$$

where $\tau$ specifies the minimum acceptable retention and $\alpha$ scales the incentive. Intuitively, $\tau$ trades off stability and ambition: too low a threshold tolerates under retention; too high makes positive signals sparse and slows learning. In practice, we set $\tau = 0.6$ as a balanced choice validated by ablations (it yields the best mean across benchmarks while maintaining stable training), and we defer full sensitivity analysis to our ablation section. To avoid rewarding confidently wrong behaviours and amplifying spurious patterns, we gate this bonus by correctness:

$$R_i = r_i + \mathbb{1}[\text{correct}] \, r_c, \tag{8}$$

So only semantically valid generations can earn retention credit. This gating reduces reward hacking, stabilises learning signals, and focuses policy updates on trajectories that already satisfy task constraints. We normalise advantages within each group to reduce variance,

$$A_i = \frac{R_i - \bar{R}}{\sigma_R}, \tag{9}$$

and optimise the clipped objective with a KL anchor to a reference policy:

$$\mathcal{L}_{\text{C-GRPO}} = \mathbb{E}\left[\frac{1}{G} \sum_{i=1}^{G} \left(\text{clip}\left(\frac{\pi_\theta(o_i \mid q)}{\pi_{\theta_{\text{old}}}(o_i \mid q)}, \ 1 - \epsilon, \ 1 + \epsilon\right) A_i\right) - \beta \, \text{KL}\big(\pi_\theta \,\|\, \pi_{\text{ref}}\big)\right]. \tag{10}$$

Together with the memory-aware compressor, C-GRPO turns compression into an alignment problem, retaining the teacher's reasoning where it matters rather than a purely geometric token reduction heuristic, yielding both efficiency and robustness under extreme temporal compression.

## 4 EXPERIMENTS

### 4.1 EXPERIMENT SETTINGS

**Benchmarks.** To evaluate the effectiveness of our method, we conduct experiments on a suite of six widely used benchmarks: VSI-Bench (Yang et al., 2025a), VideoMMMU (Hu et al., 2025), MMVU (Zhao et al., 2025), MVBench (Li et al., 2024b), TempCompass (Liu et al., 2024), and VideoMME (Fu et al., 2025). Figure 2 illustrates the evaluation benchmarks, showing the distribution based on the number of QA samples in each dataset.

**Implementation details.** For all benchmark evaluations, videos are first uniformly sampled at 1 fps, then subsampled to ensure that no more than 64 frames are processed per video. We adopt top_p = 0.001 and temperature = 0.01 to achieve greedy decoding. Flash Attention 2 (Dao, 2023) is used as the efficient attention operator. All benchmark evaluations are performed on NVIDIA A6000 GPUs with 48GB of memory. Appendix A includes more details. Our baseline experiments are conducted using Qwen2.5-VL-3B (Bai et al., 2025). We further evaluate on several widely used small-scale vLLMs, including Gemma3 (Team et al., 2025), InternVL-3.5-2B, and InternVL-3.5-4B (Wang et al., 2025). In addition, we compare our approach against representative training-free token compression strategies, namely ByteVideoLLM (Wang et al., 2024a), MovieChat (Song et al., 2024), and VidCom (Liu et al.,

| Models | Frames | Video Reasoning Benchmark | | | Video General Benchmark | | | |
| --- | --- | --- | --- | --- | --- | --- | --- | --- |
| | | VSI-Bench | VideoMMMU | MMVU (mc) | MVBench | TempCompass | VideoMME (w/o sub) | mean |
| Qwen2.5-VL-3B (Bai et al., 2025) | 64 | 32.93 | 35.33 | 48.64 | 44.77 | 38.05 | 53.55 | 42.21 |
| Qwen2.5-VL-3B (Bai et al., 2025) | 16 | 27.63 | 30.78 | 45.28 | 43.89 | 37.95 | 44.37 | 38.32 |
| InternVL3.5-2B (Wang et al., 2025) | 64 | 14.65 | 15.56 | 22.88 | 14.71 | 23.63 | 4.26 | 15.95 |
| InternVL3.5-4B (Wang et al., 2025) | 64 | 28.96 | 33.33 | 47.51 | 44.71 | 58.34 | 39.15 | 42.00 |
| Gemma-3-4B (Team et al., 2025) | 64 | 26.83 | 26.78 | 41.76 | 36.82 | 55.04 | 46 | 38.87 |
| ByteVideoLLM-3B (Wang et al., 2024a) | 64 | 21.33 | 22.33 | 28.63 | 22.56 | 35.55 | 22.7 | 25.52 |
| MovieChat-3B (Song et al., 2024) | 1 | 25.14 | 25.78 | 39.35 | 37.1 | 38.79 | 26.41 | 32.10 |
| VidCom$^2$-3B (Liu et al., 2025) | 64 | 25.5 | 23.89 | 31.08 | 29.88 | 35.23 | 21.48 | 27.84 |
| **MARC-3B** | **1** | **27.55** | **33.11** | **51.99** | **45.82** | **55.34** | **39.44** | **42.20** |

Table 1: **Performance of different models/methods on benchmarks.** We evaluated three models (Qwen2.5-VL, InternVL3.5, and Gemma-3) and three compression methods (ByteVideoLLM, MovieChat, and VidCom$^2$) using a unified set of parameters. All models and methods employ 1fps sampling, but the maximum frame rate is capped (as indicated in the frame column). *Note: The "Frame" column indicates the number of visual tokens comparable to how many frame's token that are fed into the LLM decoder before generation, not the raw video sampling rate.*

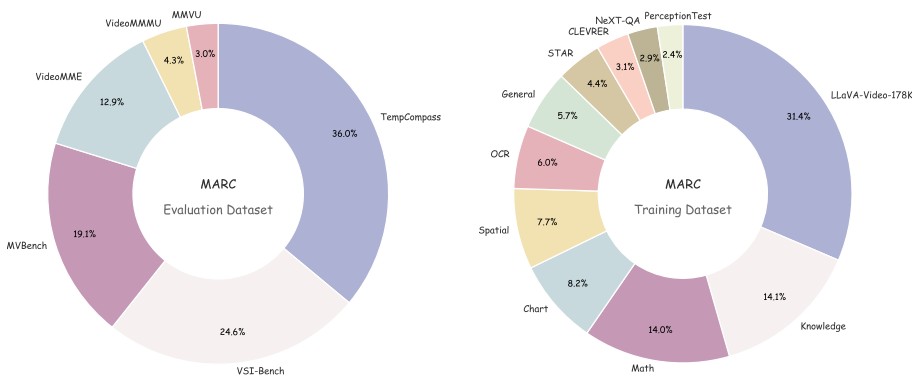

Figure 2: Distribution of benchmarks based on the number of QA samples.

Figure 3: Distribution of training dataset based on number of QA samples.

2025). We reproduce their methods on Qwen2.5-VL-3B. For temporal compression methods (e.g., MovieChat), inputs are compressed to a single frame, yielding the same effective input length as our method. For spatial or mixed compression methods (e.g., VidCom and ByteVideoLLM), we ensure that the average number of vision tokens is approximately 120, equivalent to the number in our method. Only the MARC-3B experiments employ VMR for benchmark processing, with top-$k = 3$. Further implementation details are included in Appendix A.

**Training data.** For training, we utilize the Video-R1-260K dataset (Feng et al., 2025), which is sampled from a variety of public datasets. We only randomly sampled 5K instances from this dataset, consisting of videos and images, for C-GRPO training. While the image data will not contribute to compression reward, it serves to help models develop generalized reasoning abilities in static contexts (Feng et al., 2025). The data distribution is listed in Figure 3. Appendix B contains more details regarding the training data.

**Training details**. We adopt Qwen2.5-VL-3B as the backbone model for training. The training dataset is first pre-processed using the Visual Memory Retriever. During C-GRPO training, the full-frame teacher model processes videos with 64 frames, while the student model operates on the compressed single-frame input. The ordered group size $G$ is set to 8. Additional implementation details are provided in Appendix B.

## 4.2 MAIN COMPARISON

**Performance Comparison.** Table 1 presents the results on six benchmarks, covering both video reasoning and general video understanding tasks. Before compression, the mean number of visual tokens per sample across all benchmarks (64 frames) is **2589.93**. After compression, this number is reduced to **122.69** tokens (a 95% reduction). Our method demonstrates competitive performance across all benchmarks compared with the baselines. Specifically, relative to the Qwen2.5-VL-3B baseline (64 frames), our model achieves nearly identical mean performance (42.20 vs. 42.21) while using only **4.71%** of the visual tokens, as shown in Figure 4.

Notably, the average score of MARC-3B also surpasses that of larger models such as InternVL3.5-4B and Gemma-3-4B. Among the six benchmarks, our method outperforms the Qwen2.5-VL-3B baseline on MMVU, MVBench, and Temp-Compass. The substantial improvement on TempCompass can be attributed to the enhanced instruction-following ability obtained through our training process, which effectively addresses the weakness of small-scale (3B) models.

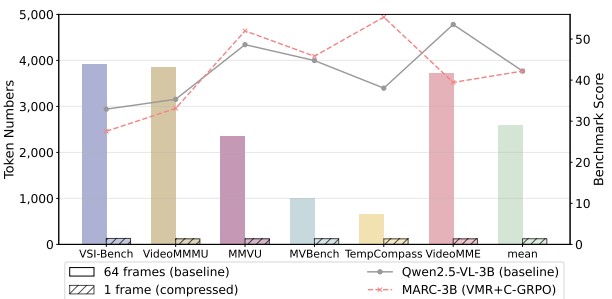

Figure 4: Vision tokens for each benchmark and MARC compared with the baseline performance.

For long video evaluation on VideoMME, our model inevitably incurs some performance loss due to extreme compression, retaining **74%** of the baseline performance while processing only **3.21%** of the original visual tokens. Nevertheless, even under this challenging setting, our approach still outperforms larger models such as InternVL3.5-4B, underscoring the effectiveness of C-GRPO in balancing efficiency and accuracy. Extensive analysis of this can be found in Appendix A.4.

Our method substantially outperforms prior compression strategies. Compared to DynamicVLM, MovieChat, and VidCom, our approach improves mean accuracy by **65%**, **31%**, and **52%**, respectively. MovieChat also employs short-term memory for temporal compression, and performs best among these methods, achieving results close to ours on VSI-Bench; however, it lags significantly on the other five benchmarks. VidCom adopts a selective token retention strategy, which has a similar effect to VMR, but still falls short of our model. These results indicate that naive compression strategies suffer from performance degradation under aggressive compression ratios.

**Efficiency Comparison.** Figure 5 reports the real-world inference efficiency of different compression strategies. We evaluate GPU peak memory usage and inference latency on MMVU multiple-choice samples using a single NVIDIA A6000 GPU. To ensure robustness of the comparison, we fix the inference prompt and set the maximum output length to generate only the answer. More details are included in Appendix A. With a batch size of 15, the baseline Qwen2.5-VL with 64 frames occupies 41.63 GB out of 48 GB of GPU memory. When applying our compression framework, the memory usage is reduced to 11.48 GB, corresponding to a **72.4%** reduction.

In terms of latency, input compression substantially reduces the number of tokens required during LLM generation, resulting in a **23.9%** reduction in generation latency. This improvement leads to a **15.9%** reduction in overall model generation latency. Consequently, the end-to-end latency for processing a single MMVU sample is reduced by **11.1%**. These results confirm that our method achieves substantial improvements in memory efficiency and inference speed, enabling faster and more resource-efficient deployment of vLLMs in practice.

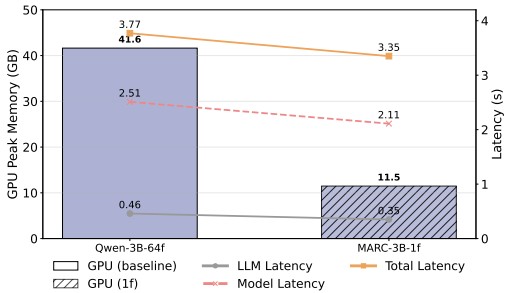

Figure 5: Efficiency Comparison: We compared GPU memory usage and LLM generation rates.

| Models | Frames | VSI-Bench | VideoMMMU | MMVU (mc) | MVBench | TempCompass | VideoMME (w/o sub) | mean |
|---|---|---|---|---|---|---|---|---|
| Qwen2.5-VL-3B(baseline) | 64 | 32.93 | 35.33 | 48.64 | 44.77 | 38.05 | 53.55 | 42.21 |
| Qwen2.5-VL-3B(VMR) | 64 | 34.02 | 34.33 | 55.52 | 57.24 | 40.38 | 51.85 | 45.56 |
| Qwen2.5-VL-3B(SFT+VMR) | 1 | 26.71 | 31.00 | 47.83 | 43.91 | 53.76 | 37.77 | 40.16 |
| Qwen2.5-VL-3B(SFT) | 1 | 25.70 | 28.33 | 45.76 | 39.48 | 54.05 | 37.72 | 38.50 |
| **MARC-3B-0.6(VMR)** | 1 | 27.55 | 33.11 | 51.99 | 45.82 | 55.34 | 39.44 | 42.20 |

Table 2: Comparative experiments evaluating the effect of the Visual Memory Retriever (VMR) and contrasting our proposed MARC distillation framework with conventional supervised fine-tuning.

| Models | $\tau$ | Frames | VSI-Bench | VideoMMMU | MMVU (mc) | MVBench | TempCompass | VideoMME (w/o sub) | mean |
|---|---|---|---|---|---|---|---|---|---|
| **MARC-3B** | 0.4 | 1 | 28.27 | 31.66 | 49.12 | 45.21 | 54.72 | 39.07 | 41.34 |
| **MARC-3B** | 0.6 | 1 | 27.55 | 33.11 | 51.99 | 45.82 | 55.34 | 39.44 | 42.20 |
| **MARC-3B** | 0.8 | 1 | 28.23 | 31.78 | 49.34 | 45.89 | 54.12 | 39.03 | 41.40 |

Table 3: Ablation study results of MARC-3B with different $\tau$.

## 4.3 ABLATION STUDIES

**In comparison to training-based compression using SFT.** To evaluate the overall effectiveness of our framework, we first train a model using standard SFT on 10K samples from Video-R1-COT-165K (Feng et al., 2025), and evaluate it on the same benchmarks (implementation details are provided in Appendix B). Table 2 shows the result. Comparing MARC-3B-0.6 with this SFT baseline, we observe consistent improvements across all benchmarks. In terms of mean performance, our model achieves 42.20 compared to 38.50, yielding a relative gain of **+9.6%**. Furthermore, even against a stronger SFT+VMR variant, our method still delivers higher scores across all benchmarks (increasing mean score by 5%. These results demonstrate that the integration of C-GRPO with VMR is crucial for boosting performance under extreme temporal compression.

**Ablation studies on the effect of $\tau$ in C-GRPO.** Table 3 presents the ablation study results for different threshold values $\tau$ during C-GRPO training. Recall from Equation 7 that $\tau$ specifies the minimum acceptable retention ratio relative to the teacher model's performance. We evaluate three values of $\tau$ (0.4, 0.6, and 0.8) using the same benchmarks and experimental setup described in Section 4.1. We observe that setting $\tau = 0.6$ achieves the best overall performance, with a mean score of 42.20 across six benchmarks. A lower threshold ($\tau = 0.4$) makes the reward constraint too lenient, resulting in insufficient incentive to retain the teacher model's full performance. Conversely, a higher threshold ($\tau = 0.8$) imposes an overly strict constraint, triggering the additional compression reward less frequently. This limits effective learning signals, restricts divergence from the baseline, and slightly reduces performance. These results suggest that a moderate $\tau$ strikes the right balance: it provides sufficient incentive for the model to preserve performance under compression while avoiding the overly conservative behaviour induced by stricter constraints.

**Ablation studies on VMR's effectiveness.** First, we conduct an experiment using Qwen2.5-VL-3B model without training and compression, and evaluate it with VMR benchmarks (top-$k = 3$). The result is shown in Table 2. It achieves a mean score of 45.56, compared to 42.21 for the baseline, demonstrating that the VMR could boost performance. For MVBench, the increase is as high as 27.85%. This is because, for videos with many clips, instead of sampling uniformly for 64 frames, we only sample in the top 3 important clips; therefore, more important frames are kept during evaluation. This could further explain MARC's effectiveness. It combines both the effect of VMR and C-GRPO, so that the model's reasoning ability rises, and we kept more important frames before compression. This combination achieves optimal performance. The effectiveness is further established by the result between SFT+VMR and SFT, we can see that the mean score increases by 4.3%. For VideoMMU and MVBench, this increase reaches more than 10%.

## 5 CONCLUSION

MARC is a memory-augmented reinforcement learning framework for efficient video understanding with high compression. It uses a Visual Memory Retriever to select key video segments and C-GRPO to distill knowledge from a 64-frame teacher model to a 1-frame student. This approach reduces visual tokens by 95% while maintaining strong performance, even outperforming the baseline on specific benchmarks. MARC is a practical solution for real-world applications in resource-constrained environments, such as real-time video question answering, surveillance, and autonomous driving.

ETHICS STATEMENT

We affirm that this research fully complies with the ICLR Code of Ethics. This study did not involve human subjects, nor did it process any datasets containing personally identifiable information or sensitive data. The methodologies and applications presented herein have no potential for harmful impacts. We have made every effort to ensure the fairness, transparency, and integrity of the research process and its outcomes. No conflicts of interest exist in this work, and all data and code will be made publicly available in accordance with ICLR's principles of openness.

REPRODUCIBILITY STATEMENT

To facilitate the reproducibility of our work, we provide detailed implementation information corresponding to the results reported in this paper. The settings for the main results (Figure 4 and Table 1) are described in Appendix A.1, Appendix A.2, and Section 4.1. For the efficiency experiments (Figure 5), implementation details are provided in Appendix A.3. A detailed description and processing steps of the training dataset (Figure 3) are included in Appendix B.1. Finally, the implementation details of C-GRPO and SFT training are presented in Appendix B.1 and Appendix B.2, respectively.

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

## A   BENCHMARK EVALUATION DETAILS

### A.1   IMPLEMENTATION

For the six benchmarks, each video is sampled at $1\,\mathrm{FPS}$ and then subsampled to a maximum of $64$ frames. Each frame is further capped to a resolution of $128\times28\times28$, ensuring that the Qwen2.5-VL visual processor produces an integer number of $14\times14$ patches. Decoding is configured to approximate greedy behavior with top_p$= 0.001$ and temperature$= 0.01$, and the maximum output length is set to $1024$ tokens.

We adopt the prompt from Video-R1 (Feng et al., 2025) for both evaluation and training. This prompt follows a Chain-of-Thought design, with additional formatting and output-type control to facilitate clearer reasoning and more reliable scoring.

**The prompt is:**

```
QUESTION_TEMPLATE = (
    "{Question}\n"
    "Please think about this question as if you were a human
    pondering deeply. "
    "Engage in an internal dialogue using expressions such as
    'let me think', 'wait', "
    "'Hmm', 'oh, I see', 'let's break it down', etc., or
    other natural language "
    "thought expressions. It's encouraged to include
    self-reflection or verification "
    "in the reasoning process. Provide your detailed
    reasoning between the <think> "
    "and </think> tags, and then give your final answer
    between the <answer> and "
    "</answer> tags."
)

TYPE_TEMPLATE = {
    "multiple choice": " Please provide only the single
    option letter (e.g., A, B, C, D, etc.) within the
    <answer> </answer> tags.",
    "numerical":       " Please provide the numerical
    value (e.g., 42 or 3.14) within the <answer>
    </answer> tags.",
    "OCR":             " Please transcribe text clearly
    and provide your answer within the <answer> </answer> tags.",
    "free-form":       " Please provide your text answer
    within the <answer> </answer> tags.",
    "regression":      " Please provide the numerical
    value (e.g., 42 or 3.14) within the <answer> </answer> tags."
}
```

Table 4 summarizes the statistics for each benchmark, reporting the average number of visual tokens per sample under three settings: 64 frames, 16 frames, and compression to a single frame. Since MVBench and TempCompass contain many short videos of only a few seconds, their average token counts are lower than those of the other benchmarks.

| Models | Frames | VSI-Bench | VideoMMMU | MMVU (mc) | MVBench | TempCompass | VideoMME (wo sub) | mean |
|--------|--------|-----------|-----------|-----------|---------|-------------|-------------------|------|
| video token number | 64 | 3917.61 | 3849.23 | 2364.87 | 1005.06 | 668.84 | 3733.96 | 2589.93 |
| video token number | 16 | 1040 | 963.70 | 937.60 | 640.28 | 609.79 | 959.82 | 858.53 |
| video token number | 1 | 130 | 120.46 | 121.51 | 124.07 | 120 | 120.08 | 122.69 |

Table 4: Average vision token numbers per sample for all six benchmarks under settings of 64, 16 and 1 frame.

## A.2 Implementation of other compression strategies

In order to compare the efficiency of our method from other compression strategies, we reproduce their method on Qwen2.5-VL-3B. VidCom (Liu et al., 2025) compresses tokens within each frame based on their importance both locally (within the frame) and globally (across frames), while maintaining a global budget on the final token count via a retention ratio. For fair comparison, we adjust the retention ratio according to Table 4, ensuring that the average number of tokens for each benchmark after VidCom compression remains around 120 for each benchmark. For ByteVideoLLM (Wang et al., 2024a), which applies spatial pooling when vision tokens pass through the Vision Transformer, we similarly adjust the pooling ratio so that the resulting token count is approximately 120. For MovieChat (Song et al., 2024), which compresses short-term memory before consolidating into long-term memory, we follow its workflow: short-term memories are first compressed to two frames, and then further compressed into a single long-term memory frame. The resulting token count is thus aligned with our method. These adjustments ensure that all baselines operate under a comparable token budget, making the evaluation fair and consistent.

## A.3 Implementation of efficiency experiment

To evaluate the efficiency of MARC, we fix the prompt template such that the model outputs only the final answer to ensure a fair comparison of inference cost. Evaluation is conducted on MMVU samples with a batch size of 15 in order to maximize GPU memory utilization. We use 1 NVIDIA A6000 GPU to evaluate. All other evaluation settings are kept identical to those used in the performance experiments.

## A.4 Analysis on failure case

From Table 2, comparing Qwen2.5-VL-3B (baseline) and Qwen2.5-VL-3B (VMR), we observe that VideoMME does not benefit from VMR, with the score dropping from 53.55 to 51.85. A key factor lies in the nature of the benchmark. VideoMME contains significantly longer videos: one third are of medium length with an average of 515 seconds, and another third are long videos averaging 2466 seconds. By contrast, the longest among the other benchmarks, VideoMMMU, averages 506 seconds. Notably, VideoMMMU also shows a slight drop under VMR. Since longer videos are divided into more clips, restricting retrieval to the top-3 inevitably discards critical information, whereas the four shorter benchmarks, with fewer clips and shorter durations, benefit consistently from VMR. Attempts to increase from top 3 to more clips doesn't simply solve this, because for longer videos the number of clips could reach more than 100.

MARC, despite distilling stronger video understanding abilities from the 64-frame teacher, still falls short on very long videos compared to short ones, as it cannot recover information once most of the temporal chain has been discarded. As a result, it retains only 74% of baseline performance on VideoMME. This is consistent with the VideoMME paper, which reports that increased sparsity in frame sampling reduces effective input information (Fu et al., 2025).

A potential mitigation strategy is to incorporate a temporal Q-Former that adaptively identifies key frames for compression, which could be further explored.

Overall, these findings suggest that our compression framework works very well for short clips (MMVU, MVBench, TempCompass), performs reasonably on medium-length videos (VSI-Bench, VideoMMMU), but sacrifices some performance on long-duration videos (VideoMME).

## B Training Details

### B.1 More details of C-GRPO

The 5K training dataset, illustrated in Figure 3, contains both video and image sources. Apart from the specific video datasets shown, the composition of the categories in the figure is as follows:

- **Knowledge:** TQA, AI2D, ScienceQA, PMC-VQA, VQA-RAD, GVLQA, ArxivQA, EXAMS-V, AI2D-gpt4v.

| Method | MARC (1 frame) | MARC (8 frames) | Qwen (16 frames) | Qwen (64 frames) |
|---|---|---|---|---|
| VideoMME Result | 39.44 | 45.33 | 44.37 | 53.55 |

Table 5: Performance comparison on the VideoMME benchmark under different visual token budgets. MARC maintains competitive performance under aggressive compression while using significantly fewer visual tokens compared to native multi-frame baselines.

- **Math:** CLEVR-Math, GEOS, Geometry3K, GeoQA+, UniGeo, Multimath-300K, Super-CLEVR.
- **Chart:** FigureQA, DVQA, PlotQA, ChartQA, MapQA, TabMWP, Chart2Text, RoBUT, SQA, VisualWebInstruct.
- **Spatial:** OpenSpaces, Spacellava.
- **OCR:** TextVQA, HME100k, ChromeWriting, IAM, Rendered Text, TextCaps, TextOCR.
- **General:** A-OKVQA, IconQA, ShareGPT4V, Visual7W, ShareGPT4o.

This 5K training dataset is sampled from the Video-R1-260K dataset (Feng et al., 2025), preserving the same category proportions. After sampling, we apply VMR to process the 5K dataset, retaining only the top-3 most important clips in each video. This is supported by Table 2, which shows that applying VMR generally leads to improved performance.

Qwen2.5-VL-3B serves as the backbone model for our C-GRPO training. Each frame is capped at a maximum resolution of $128 \times 28 \times 28$ pixels. The ordered group size is set to $G = 8$, and the maximum completion length is limited to 768 tokens. We adopt the Adam optimizer with a learning rate of $1 \times 10^{-6}$ and a weight decay of 0.01. We set $\beta$, the KL divergence term of the GRPO objective, to 0.04. DeepSpeed ZeRO-3 is used for memory-efficient distributed optimization.

### B.2 Details of training-based compression using SFT

For SFT, we sample 10K instances from the Video-R1-COT-165K dataset (Feng et al., 2025), which is constructed by leveraging Qwen2.5-VL-72B-Instruct (Bai et al., 2025) to generate chain-of-thought (CoT) rationales for samples in Video-R1-260K, followed by filtering.

We then fine-tune the Qwen2.5-VL-3B-Instruct model using SFT. Training is conducted on 4 NVIDIA H100 GPUs with the Adam optimizer, using a learning rate of $1 \times 10^{-6}$ and gradient clipping at 5. We use bfloat16, and employ DeepSpeed ZeRO-2.

## C The Use of Large Language Models

Large Language Models (LLMs) were employed solely to assist with polishing the writing, such as improving grammar and refining wording. They were not used for ideation or for generating original content, and no sections of text were produced purely by LLMs.

## D Additional Results in ICLR 2026 Rebuttal

### D.1 Effectiveness of Adjustable Compression and Clarification on Performance Trade-offs

Table 5 presents additional results on the VideoMME benchmark under different compression ratios. To further analyse the impact of visual token allocation, we evaluate MARC under both single-frame equivalent compression and reduced compression (8-frame equivalent compression), and compare with native multi-frame baselines.

Under single-frame equivalent compression, MARC achieves a score of 39.44 on VideoMME. It is worth noting that VideoMME contains substantially longer videos compared to typical benchmarks. Approximately one third of the videos fall into the medium-length regime (average 515 seconds), while another third have an average duration of approximately 2,466 seconds. Under such extreme

| Frame Configuration | Model Generation Latency (s) | | | Token Number | |
|---|---|---|---|---|---|
| | Before | Compressed | Decrease | Before | Compressed |
| 512 frames, $256 \times 28 \times 28$ pixels/frame | 18.36 | 7.65 | 58.30% | 64,512 | 1,008 |
| 256 frames, $256 \times 28 \times 28$ pixels/frame | 8.26 | 4.55 | 44.90% | 32,256 | 504 |
| 128 frames, $128 \times 28 \times 28$ pixels/frame | 2.80 | 2.29 | 18.20% | 7,689 | 120 |
| 64 frames, $128 \times 28 \times 28$ pixels/frame (ours) | 2.51 | 2.11 | 15.90% | 3,840 | 120 |

Table 6: Inference latency comparison under a fixed compression ratio (retaining approximately 5% of visual tokens) across different input scales. The acceleration benefit becomes more pronounced as the total visual token count increases.

temporal sparsity, aggressive compression inevitably removes information that cannot be fully recovered.

When the compression ratio is reduced to 8-frame equivalent compression, performance improves to 45.33 while still using only 12.8% of the visual tokens. In comparison, the native Qwen model achieves 44.37 points at 16 frames and 53.55 points at 64 frames. These results indicate that the RL distillation framework effectively improves performance when moderate visual token budgets are allocated, while maintaining substantially lower token cost compared to native baselines.

Overall, the observed performance differences reflect a controllable trade-off between token budget allocation and temporal information preservation, rather than intrinsic limitations of the compression framework itself.

These findings further suggest a natural extension toward dynamic compression strategies. In particular, token budgets could be adaptively allocated based on video duration or content complexity to better preserve critical temporal dependencies. Exploring adaptive token allocation mechanisms remains an important direction for future work.

### D.2 LATENCY IMPROVEMENTS DEPEND ON TOKEN SCALE

Table 6 provides additional analysis of inference latency under a fixed compression ratio (approximately 5% visual token retention) across different input scales. We measure model generation latency before and after compression to study the relationship between token count and latency reduction.

In high-token regimes (e.g., 512 frames corresponding to 64,512 tokens), compression reduces inference latency from 18.36 seconds to 7.65 seconds, corresponding to a 58.3% speedup. Similarly, in the 256-frame setting, compression achieves a 44.9% latency reduction. As the total token count decreases, the relative acceleration also decreases, with only 15.9% speedup observed in the 64-frame setting.

This behaviour can be explained by the transition between different system bottleneck regimes. When the number of uncompressed input tokens is small, the system operates outside the visual-token bottleneck regime, and compression provides limited relative speedup. In contrast, when the input token count is large, inference latency becomes increasingly dominated by LLM decoding and attention computation costs. In this regime, reducing visual token count leads to substantial latency reduction.

It is also worth noting that the relatively modest token counts used in our main experiments are primarily determined by training-time computational constraints, including frame number and per-frame spatial resolution. In real-world long-video scenarios, where visual token counts can be significantly larger, compression-based methods are expected to provide more substantial acceleration benefits.

Overall, these results demonstrate that the efficiency gain of the proposed framework scales favourably with input token count, making it particularly suitable for long-video understanding scenarios with high visual token budgets.

