# OpenReview forum: "MARC: Memory-Augmented RL Token Compression for Efficient Video Understanding"
_ICLR.cc/2026/Conference — ICLR 2026 Poster_

### Official Review · Reviewer_CzmY · 2025-10-28

**Soundness:** 3
**Presentation:** 3
**Contribution:** 3
**Rating:** 6
**Confidence:** 4

**Summary:**

This paper proposes MARC, a retrieve-then-compress framework for efficient video understanding. The authors introduce the Compression Group Relative Policy Optimization (C-GRPO) algorithm, which transfers the complex reasoning ability from a high-token "Teacher" network to a low-token "Student" network. The method achieves performance comparable to full-token models while drastically reducing GPU memory consumption.

**Strengths:**

1. The proposed method is effective, successfully achieving performance with only 1 token that rivals models using 64 tokens.
2. The introduced C-GRPO algorithm is interesting and demonstrates successful transfer of complex reasoning capabilities from a high-token "Teacher" network to a low-token "Student" network.

**Weaknesses:**

1. MARC appears to have significant limitations in long video understanding. Has the method been evaluated under scenarios where a high compression ratio is maintained but the input token count is increased to better accommodate longer videos?
2. While the reduction in the number of tokens is impressive, the end-to-end latency improvement seems relatively limited.

**Questions:**

Related to the weaknesses above, have you explored strategies to better adapt MARC to long video inputs, possibly by increasing the input token count while maintaining a high compression ratio?

---

> ### Author Response · Authors · 2025-11-17
> **Effectiveness of Adjustable Compression and Clarification on Performance Trade-offs**
>
> |                           | MARC (1frame) | MARC (8frames) | Qwen (16frames) | Qwen (64frames) |
> |----------------------------------|-------------|-------------|-------------|--------------|
> | **VideoMME Result**                       | 39.44       | 45.33       | 44.37       | 53.55        |
>
>
> We are most grateful to the reviewer for your insightful observations regarding the model's performance degradation on extremely long videos. Following your kind advice to maintaining high compression ratio with increased input token count, we have supplemented our experimental results by reducing the compression rate (from 1-frame equivalent compression to 8-frame equivalent compression) to validate our method’s effectiveness. Specifically, on the VideoMME benchmark, our approach achieved a score of 39.44 under single-frame equivalent compression. When the compression rate was reduced to 8-frame equivalent compression, the score significantly improved to 45.33 (with 12.8% tokens). By contrast, the native Qwen achieved 44.37 points at 16 frames and reached 53.55 points at 64 frames. These results demonstrate that our RL distillation framework can effectively improve performance by appropriately reducing compression ratios while maintaining a significantly lower visual token cost than the baseline. This proves that the decline in model performance stems not from framework limitations, but from a controllable trade-off in token budget allocation. Furthermore, this demonstrates our design's potential for natural extension to dynamic compression strategies: adaptively allocating token budgets based on video length or content complexity to more effectively preserve critical temporal link information (an area we intend to explore as a future research focus). We extend our sincere gratitude once more to the reviewers for their constructive feedback, which has been invaluable in refining both our methodology and manuscript.

---

> ### Author Response · Authors · 2025-11-17
> **Latency Improvements Depend on Token Scale**
>
> | Frame Configuration | Model Generation Latency (s) |  |  | Token number |  |
> |----------------------|-------------------------------|--|--|--------------|--|
> |                      | **Before** | **Compressed** | **Decrease** | **Before** | **Compressed** |
> | 512 frame, 256×28×28 pixels/frame | 18.36 | 7.65 | 58.30% | 64,512 | 1,008 |
> | 256 frame, 256×28×28 pixels/frame | 8.26 | 4.55 | 44.90% | 32,256 | 504 |
> | 128 frame, 128×28×28 pixels/frame | 2.80 | 2.29 | 18.20% | 7,689 | 120 |
> | 64 frame, 128×28×28 pixels/frame *(our paper)* | 2.51 | 2.11 | 15.90% | 3,840 | 120 |
>
> We appreciate the reviewers' attention to the issue of inference efficiency. We have conducted supplementary experiments to further investigate the latency behavior and found that, under the same compression ratio, the performance gain becomes more pronounced as the LLM input token count increases.
>
> Using a fixed compression ratio (retaining 5% of visual tokens), we measured model generation latency before and after compression across a range of visual input scales. As shown in the table, in high-token scenarios (e.g., 512 frames, 64,512 tokens), MARC reduces inference latency from 18.36 s to 7.65 s, achieving a 58.3% acceleration, and achieves 44.9% acceleration in the 256-frame scenario. As the token count decreases, the acceleration rate diminishes accordingly (e.g., only 15.9% acceleration for 64 frames). This occurs because when the number of uncompressed input tokens is small, the system operates outside the visual-bottleneck regime, so compression provides only limited relative speedup. In contrast, when the input token count is large, latency becomes dominated by LLM decoding speed, and higher compression ratios yield substantial performance gains. Therefore, these findings do not indicate any inherent efficiency disadvantage in the framework.
>
> The latency issue raised by the reviewer arises under our experimental setting, where the token counts are relatively modest due to the limited frame numbers and per-frame resolution we adopted to fit within our training-time computational budget. Overall, our method could provide significant acceleration benefits for real-world long-video scenarios with high token counts. We extend our gratitude once more to the reviewers for raising these points, which enabled us to supplement our experimental validation with more comprehensive testing.

---

### Official Review · Reviewer_VGv8 · 2025-10-31

**Soundness:** 3
**Presentation:** 2
**Contribution:** 3
**Rating:** 6
**Confidence:** 4

**Summary:**

This work presents MARC, a new framework for efficiently understanding videos. First, MARC segments videos into event-level clips. Then, it employs a Visual Memory Retriever to select relevant video segments, which are compressed using a Reinforcement Learning-based Compression Group Relative Policy Optimization (C-GRPO) strategy to reduce tokens. C-GRPO distills the Teacher Network, which takes 64 frames as input, into the Student Network, which takes just one frame's worth of tokens as input. This significantly reduces peak GPU memory usage while achieving comparable performance in some benchmarks (41.6G $\rightarrow$ 11.5G).

The novelty of this work lies in the C-GRPO, which is the first method to compress video tokens using reinforcement learning. Extensive experiments prove its obvious advantage over other token compression methods, despite the extra training required. I believe this will pave the way for a new approach to token compression for video understanding.

However, I still hold some concerns:
1. Given the significant reduction in visual tokens, why is the inference latency nearly identical to that of the baseline? What is the main bottleneck limiting inference efficiency?
2. As shown in Table 1, the performance gap between MARC and the baseline (Qwen-VL 2.5) varies dramatically across benchmarks—MARC either substantially outperforms or noticeably underperforms the baseline. A more detailed analysis of these discrepancies across different benchmarks would be helpful.

Overall, I like this work and advocate its acceptance if my concerns are properly answered.

**Strengths:**

This work presents MARC, a new framework for efficiently understanding videos. First, MARC segments videos into event-level clips. Then, it employs a Visual Memory Retriever to select relevant video segments, which are compressed using a Reinforcement Learning-based Compression Group Relative Policy Optimization (C-GRPO) strategy to reduce tokens. C-GRPO distills the Teacher Network, which takes 64 frames as input, into the Student Network, which takes just one frame's worth of tokens as input. This significantly reduces peak GPU memory usage while achieving comparable performance in some benchmarks (41.6G $\rightarrow$ 11.5G).

The novelty of this work lies in the C-GRPO, which is the first method to compress video tokens using reinforcement learning. Extensive experiments prove its obvious advantage over other token compression methods, despite the extra training required. I believe this will pave the way for a new approach to token compression for video understanding.

**Weaknesses:**

1. Given the significant reduction in visual tokens, the inference latency is nearly identical to that of the baseline, indicating there is a bottleneck limiting inference efficiency
2. As shown in Table 1, the performance gap between MARC and the baseline (Qwen-VL 2.5) varies dramatically across benchmarks—MARC either substantially outperforms or noticeably underperforms the baseline. A more detailed analysis of these discrepancies across different benchmarks would be helpful.
3. The overview figures (Figure 1 and Figure 2) lack essential annotations and explanatory elements, making them difficult to interpret. I recommend enriching the figure with additional clarifying details to help readers better understand the proposed method.

**Questions:**

1. Given the significant reduction in visual tokens, why is the inference latency nearly identical to that of the baseline? What is the main bottleneck limiting inference efficiency?
2. As shown in Table 1, the performance gap between MARC and the baseline (Qwen-VL 2.5) varies dramatically across benchmarks—MARC either substantially outperforms or noticeably underperforms the baseline. A more detailed analysis of these discrepancies across different benchmarks would be helpful.

---

> ### Author Response · Authors · 2025-11-17
> **1. Latency Improvements Depend on Token Scale**
>
> | Frame Configuration | Model Generation Latency (s) |  |  | Token number |  |
> |----------------------|-------------------------------|--|--|--------------|--|
> |                      | **Before** | **Compressed** | **Decrease** | **Before** | **Compressed** |
> | 512 frame, 256×28×28 pixels/frame | 18.36 | 7.65 | 58.30% | 64,512 | 1,008 |
> | 256 frame, 256×28×28 pixels/frame | 8.26 | 4.55 | 44.90% | 32,256 | 504 |
> | 128 frame, 128×28×28 pixels/frame | 2.80 | 2.29 | 18.20% | 7,689 | 120 |
> | 64 frame, 128×28×28 pixels/frame *(our paper)* | 2.51 | 2.11 | 15.90% | 3,840 | 120 |
>
> We appreciate the reviewers' attention to the issue of inference efficiency. We have conducted supplementary experiments to further investigate the latency behavior and found that, under the same compression ratio, the performance gain becomes more pronounced as the LLM input token count increases.
>
> Using a fixed compression ratio (retaining 5% of visual tokens), we measured model generation latency before and after compression across a range of visual input scales. As shown in the table, in high-token scenarios (e.g., 512 frames, 64,512 tokens), MARC reduces inference latency from 18.36 s to 7.65 s, achieving a 58.3% acceleration, and achieves 44.9% acceleration in the 256-frame scenario. As the token count decreases, the acceleration rate diminishes accordingly (e.g., only 15.9% acceleration for 64 frames). This occurs because when the number of uncompressed input tokens is small, the system operates outside the visual-bottleneck regime, so compression provides only limited relative speedup. In contrast, when the input token count is large, latency becomes dominated by LLM decoding speed, and higher compression ratios yield substantial performance gains. Therefore, these findings do not indicate any inherent efficiency disadvantage in the framework.
>
> The latency issue raised by the reviewer arises under our experimental setting, where the token counts are relatively modest due to the limited frame numbers and per-frame resolution we adopted to fit within our training-time computational budget. Overall, our method could provide significant acceleration benefits for real-world long-video scenarios with high token counts. We extend our gratitude once more to the reviewers for raising these points, which enabled us to supplement our experimental validation with more comprehensive testing.

---

> ### Author Response · Authors · 2025-11-17
> **2. Why MARC Performs Differently Across Benchmarks**
>
> |                           | MARC (1frame) | MARC (8frames) | Qwen (16frames) | Qwen (64frames) |
> |----------------------------------|-------------|-------------|-------------|--------------|
> | **VideoMME Result**                       | 39.44       | 45.33       | 44.37       | 53.55        |
>
>
> As MARC operates as a temporal compression mechanism, the performance gap between MARC and the baseline varies primarily with the duration characteristics of different benchmarks. For short video benchmarks, such as MMVU (avg. 51.4s), MVBench (avg. 16s), and TempCompass (avg. 11s), MARC consistently outperforms the baseline. For medium-length datasets, such as VideoMMMU (avg. 506.2s) and VSI-Bench (avg. ~150s), MARC yields slightly lower results, retaining 88.9% of baseline performance on average.
> For VideoMME, and extra long video benchmark, the retention drops to 74% of baseline performance. This behavior is expected: VideoMME contains substantially longer videos, where one third are medium-length range (515s), and another third average in 2,466 seconds. Under such extreme temporal sparsity, aggressive compression inevitably removes information that cannot be fully recovered. Under 1-frame equivalent compression, MARC achieves a score of 39.44. When we relax compression to an 8-frame equivalent, performance improves significantly to 45.33 (85% of 64 frame-baseline). For comparison, Qwen baseline achieves 44.37 at 16 frames and 53.55 at 64 frames. This proves that the decline in model performance stems not from framework limitations, but from a controllable trade-off in token budget allocation. Furthermore, this demonstrates our design's potential for natural extension to dynamic compression strategies: adaptively allocating token budgets based on video length or content complexity to more effectively preserve critical temporal link information (an area we intend to explore as a future research focus).

---

> ### Author Response · Authors · 2025-11-17
> **3. Figures change**
>
> We are most grateful for your feedback regarding the lack of clarity in our figure descriptions, and we have made the adjustments in some way. Furthermore, the updated version of the paper incorporates additional explanatory annotations.

---

### Official Review · Reviewer_CX8T · 2025-10-31

**Soundness:** 2
**Presentation:** 2
**Contribution:** 2
**Rating:** 2
**Confidence:** 5

**Summary:**

This paper proposes a "retrieval-first then compression" framework called MARC to address the significant computational cost and latency challenges faced by video language models (VLMs) when processing long videos. The framework first uses a "visual memory retriever" (VMR) to segment the video into event-based clips and identify the most relevant clips for the user's query. It then employs a novel reinforcement learning-based distillation strategy called "Compressed Group Relative Policy Optimization" (C-GRPO) to distill the inference capability of a "teacher model" that processes 64 frames into a "student model" that uses only the token count equivalent to one frame.

**Strengths:**

Achieve a 95% visual token compression rate with almost no loss of model average accuracy, and significantly reduce memory and latency.
Originality: Core Contribution C-GRPO performs a clever "hack" on the existing GRPO algorithm by designing a new "performance retention reward" (retention alignment reward) specifically for compression tasks.

**Weaknesses:**

1.The paper lacks some comparisons with token compression works such as visionzip, videoxl, etc., and the author used some data and content from video-r1, but why there is no comparison with video-r, making the results less persuasive.
2. The motivation introduced in the intro is not strong. The author mentions that previous work caused significant information loss due to being training-free. First, can this conclusion be determined with 100% certainty? For example, if the compression ratio of training-free methods is not that large, would there still be substantial information loss? The essence of this information loss is not necessarily caused by being training-free. Second, previous methods are not all training-free; many can be trained and fine-tuned. The author introduces the use of RL training in this work, but the motivation is not very strong. For instance, why use RL and not the previously mentioned trainable methods?
3. The author can provide some use cases at the end of the main text or in the appendix to help readers better understand.

**Questions:**

1.MARC writes in the implementation part that 1 fps sampling is used with a maximum of 64 frames, but the final evaluation results table shows 1 frame. Generally, the maximum input frame number is used, not the compressed frame number. The author can check the previous token compression work paper.

---

> ### Author Response · Authors · 2025-11-17
> **1. Justification of Baseline Choices and Additional Comparisons**
>
> |       | VSIBench | videommmu | mmvu  | mvbench | tempcompass | videomme | Mean       |
> |-------------|-----------|-----------|-------|----------|--------------|-----------|--------------|
> | **MARC**     | 27.55     | 33.11     | 51.99 | 45.82    | 55.34        | 39.44     | **42.20**     |
> | **VisionZip** | 25.66     | 25.77     | 43.99 | 38.88    | 41.38        | 40.81     | **36.08**     |
>
>
> We are grateful for the reviewers' suggestions. To address your request for reproducible results for the VisionZip method, we have supplemented a comparison experiment against VisionZip across the 6 benchmarks. The results are presented in the table above.
> As demonstrated, MARC significantly outperforms VisionZip except for VideoMME, achieving an average gain exceeding six percentage points. This confirms that our RL distillation compression framework is not only viable but also delivers markedly superior performance while maintaining exceptionally low token costs. Regarding the VideoXL method mentioned by the reviewers, we must emphasise: VideoXL constitutes an inference acceleration technique for cross-layer key-value compression within the LLM itself, whereas MARC and VisionZip represent visual token compression methods operating between the vision encoder and projector. These are fundamentally distinct approaches belonging to entirely different research domains, making direct comparison inappropriate. MARC concentrates on visual token representation compression. Therefore, it is inappropriate to demand a direct comparison between the two.
>
> In addition, we contend that video-r1 is not a compression model, nor does it propose a compression method. We merely utilise the training dataset curated by video-r1, owing to its extensive coverage and high-quality data collection. Consequently, video-r1 cannot serve as one of our comparative baselines for compression discussions. We have also conducted exhaustive ablation experiments regarding the gains derived from the data, as demonstrated in Section 4.3 (lines 447–454) and Table 2. The experimental results indicate that, under identical training data conditions, our RL compression distillation framework achieves a 5% performance advantage over SFT. (SFT even utilised twice the additional data compared to RL.)

---

> ### Author Response · Authors · 2025-11-17
> **2. Justifying Our Compression Motivation and Choice of RL**
>
> 1. Firstly, we believe that achieving a high compression ratio while preserving the performance solely through training-free methods is insufficient (as demonstrated in Table 1 and Table 2).
> 2. It is widely acknowledged that higher compression ratios result in greater performance degradation, a point well illustrated by the results for VisionZip, as you mentioned earlier.
> 3. Given these two considerations, researchers are actively exploring how to achieve a balance. Consequently, our investigation into maintaining near-original performance under extreme compression conditions is of significant importance.
> 4. Conventional non training free method for compression is supervised finetuning (SFT). As discussed in Section 4.3 (lines 447–454) and reported in Table 2. We conducted extensive ablation studies on SFT based compression training, and the results consistently show that our RL-based distillation framework is more effective. Under identical training data conditions, the proposed RL compression distillation achieves an average performance gain of approximately 5% over SFT, demonstrating its clear advantage.

---

> > ### Comment · Reviewer_CX8T · 2025-11-27
> >
> > Most of my concerns have been resolved. However, I have one remaining question regarding training efficiency. Could you provide a comparison between SFT and RL in terms of computational cost (e.g., GPU hours)?
> >
> > Following our discussion and this final clarification, I am inclined to raise my score for this paper.

---

> > > ### Author Response · Authors · 2025-11-27
> > >
> > > Dear Reviewer CX8T,
> > >
> > > Thank you for your insightful feedback and for your willingness to reconsider the score. We appreciate you pointing out the specific mechanism of Qwen-2.5-VL and asking for the efficiency comparison.
> > >
> > > **1. Training Efficiency**
> > >
> > > SFT: Approximately 6 GPU hours.
> > >
> > > RL: Approximately 20 GPU hours.
> > >
> > > The RL stage incurs higher costs primarily due to the overhead of online sampling (rollouts) and reward calculation.
> > >
> > > To mitigate this, we utilized vLLM to accelerate the generation phase, keeping the training time efficient. We believe this additional cost is well justified by the significant performance gains observed.
> > >
> > > **2. Clarification on "1 Frame" Input**
> > >
> > > We fully agree that distinguishing between raw input and the compressed representation in Qwen-2.5-VL is essential to avoid ambiguity. As suggested, we have revised the caption in the manuscript to explicitly define "1 Frame" as the compressed token count processed by the LLM, ensuring a fair and holistic comparison.
> > >
> > > We hope these clarifications address your concerns. We respectfully ask if you would consider raising your score based on these responses.
> > >
> > > Best regards,
> > >
> > > The Authors

---

> ### Author Response · Authors · 2025-11-17
> **3. Use cases**
>
> As noted in the conclusion of our abstract, this approach is broadly applicable across multiple real-time scenarios, including video question answering in security edge-devices, surveillance systems, and autonomous driving. These use cases highlight the practical relevance of our method, especially for deployment on embedded devices with strict edge-side constraints.

---

> ### Author Response · Authors · 2025-11-17
> **4. Addressing the Misinterpretation of “1 Frame” in Our Evaluation Table**
>
> I believe there may be a misunderstanding. Our discussion refers specifically to the number of video tokens before they are fed into the LLM. In the baseline setting, 64 frames produce 64 frames’ worth of tokens prior to entering the LLM, whereas in our method, only a single frame worth’s tokens are passed forward. This distinction is intended solely to illustrate the compression ratio before LLM input. We apologise for any confusion. Our intention was not to suggest any difference in the underlying video frame extraction process.

---

> > ### Comment · Reviewer_CX8T · 2025-11-27
> >
> > Thank you for the detailed response. However, I would like to point out that Qwen-2.5-VL also employs a pooling mechanism after the vision encoder for compression. In your paper, the claim focuses specifically on the token count before the LLM input. This creates a potential ambiguity in the comparison.
> >
> > I suggest adding an annotation or footnote to clearly define what is meant by '1 token input' in this context. Since readers typically evaluate model architectures holistically, the term '1 token' might be misinterpreted as referring to the raw input capacity, rather than the compressed representation fed into the LLM.

---

> ### Author Response · Authors · 2025-11-25
>
> Dear Reviewer CX8T,
>
> We sincerely appreciate the time and effort you have devoted to reviewing our work.
>
> We are writing to inquire whether our previous responses have adequately addressed your concerns. If you have any further questions or suggestions, please let us know; we are eager to make the best use of the remaining time to provide further clarifications.
>
> If our responses have resolved your concerns, we would greatly appreciate it if you could reconsider your evaluation of our work.
>
> We value your feedback and look forward to hearing from you.
>
> Best regards,
>
> The Authors

---

> ### Author Response · Authors · 2025-11-26
> **Follow-up on our response to Reviewer CX8T**
>
> Dear Reviewer CX8T,
>
> Thank you again for your time and the constructive feedback provided so far.
>
> With the author-reviewer discussion period concluding on December 2nd, we are writing to kindly inquire if our previous responses have adequately addressed your concerns.
>
> We are very eager to utilize the days remaining before the deadline to provide any further clarifications or additional data you might need. If our response has resolved your concerns, we would be grateful if you could consider updating your evaluation to reflect the improvements.
>
> Best regards,
>
> The Authors

---

### Official Review · Reviewer_n2Ge · 2025-11-08

**Soundness:** 4
**Presentation:** 4
**Contribution:** 3
**Rating:** 8
**Confidence:** 4

**Summary:**

The MARC framework addresses the significant computational overhead and memory bottlenecks faced by Visual Language Models (VLMs) when processing long-duration videos, aiming to overcome the performance degradation common in prior training-free compression methods.

The MARC framework successfully solves the VLM computational bottleneck by achieving robust performance retention despite an extreme 95% reduction in visual tokens, driven by the innovative VMR and C-GRPO components. It can be a natural extension of various token compression or frame selection based on RL.

However, as all of its preceeding works, while highly efficient and accurate across most tasks, its fixed retrieval limit causes notable performance sacrifice in scenarios involving very long, complex temporal contexts. Given its substantial practical contributions to VLM deployment efficiency, the overall conclusion is a **accept**.

**Strengths:**

*   **Exceptional Efficiency and Resource Optimization:** MARC achieves a **95% reduction in visual tokens** by compressing input to the equivalent of a single frame. This results in practical gains, including a **72% reduction in GPU peak memory usage** and a **23.9% reduction in generation latency**, enabling deployment in resource-constrained, real-time applications.
*   **Performance Maintenance at Extreme Compression:** The framework maintains **nearly identical mean accuracy** (42.20 vs. 42.21) compared to the 64-frame baseline and surpasses the baseline on several complex benchmarks (MMVU, MVBench, TempCompass). It significantly outperforms prior training-free compression strategies.
*   **Novel Reinforcement Learning for Distillation:** C-GRPO ensures performance robustness under aggressive compression by using a specialized **retention alignment reward** to explicitly match the student network's output to the teacher's reasoning quality.
*   **Structured Retrieval:** The VMR segments video into semantically coherent events, confirming that this approach eliminates redundant information and even boosts baseline accuracy by focusing on query-relevant context before compression.

**Weaknesses:**

*   **Performance Degradation on Extremely Long Videos:** The model incurs performance loss when dealing with videos of average duration exceeding several hundred seconds (e.g., VideoMME), retaining only **74% of the baseline score**. This occurs because the model cannot recover information once most of the temporal chain has been discarded.
*   **Static Compression Strategy:** The framework’s commitment to a **fixed, single-frame equivalent token budget** lacks dynamic adaptability. This static approach struggles when crucial temporal context is scattered across many fragments, as the VMR's restricted retrieval inevitably discards necessary information.

**Questions:**

See summary for my main concern.

---

> ### Author Response · Authors · 2025-11-17
> **Effectiveness of Adjustable Compression and Clarification on Performance Trade-offs**
>
> |                           | MARC (1frame) | MARC (8frames) | Qwen (16frames) | Qwen (64frames) |
> |----------------------------------|-------------|-------------|-------------|--------------|
> | **VideoMME Result**                       | 39.44       | 45.33       | 44.37       | 53.55        |
>
>
>
> We are most grateful to the reviewer for your insightful observations regarding the model's performance degradation on extremely long videos and the limitations of static compression strategies. In response to the concerns raised, we have supplemented our experimental results by reducing the compression rate (from 1-frame equivalent compression to 8-frame equivalent compression) to validate our method’s effectiveness. Specifically, on the VideoMME benchmark, our approach achieved a score of 39.44 under single-frame equivalent compression. VideoMME contains substantially longer videos, where one third are medium-length range (515s), and another third average in 2,466 seconds. Under such extreme temporal sparsity, aggressive compression inevitably removes information that cannot be fully recovered. When the compression rate was reduced to 8-frame equivalent compression, the score significantly improved to 45.33 (with 12.8% tokens). By contrast, the native Qwen achieved 44.37 points at 16 frames and reached 53.55 points at 64 frames. These results demonstrate that our RL distillation framework can effectively improve performance by appropriately reducing compression ratios while maintaining a significantly lower visual token cost than the baseline. This proves that the decline in model performance stems not from framework limitations, but from a controllable trade-off in token budget allocation. Furthermore, this demonstrates our design's potential for natural extension to dynamic compression strategies: adaptively allocating token budgets based on video length or content complexity to more effectively preserve critical temporal link information (an area we intend to explore as a future research focus). We extend our sincere gratitude once more to the reviewers for their constructive feedback, which has been invaluable in refining both our methodology and manuscript.

---

### Author Response · Authors · 2025-11-21
**General Response**

Dear PCs, SACs, ACs, and Reviewers,

We sincerely appreciate the reviewers’ time, effort, and thoughtful feedback on our work. We are truly grateful for the insightful questions raised, especially those concerning compression ratios and latency. In response, we have carefully expanded our experimental results to address these points in detail. We have also updated the relevant figure as requested, with all modifications clearly highlighted in blue in the revised PDF. We look forward to any further comments or suggestions from the reviewers.

---

### Author Response · Authors · 2025-11-29
**Summary of Rebuttal**

Dear Area Chair and Reviewers,

Thank you sincerely for your time and expertise during this review cycle. We deeply appreciate your thorough work in overseeing the discussion and providing constructive feedback.

Throughout the rebuttal period, we made every effort to actively engage with and address all reviewer concerns. The quality of our paper has been strengthened by your invaluable suggestions. We are particularly encouraged by the positive reception from one of the reviewers following the discussion.

Our key modifications and clarifications, responding to each set of feedback, are summarized below:

Reviewer **n2Ge (8 points)**: This reviewer identified performance degradation on extremely long videos and the static nature of the compression strategy. We directly addressed these by introducing new, adjustable compression experiments. We demonstrated that increasing the compression budget from the default 1-frame to 8-frames significantly improves performance on VideoMME. This decisively confirms that the observed degradation is a controllable token budget trade-off, not a fundamental limitation of our framework.

Reviewer **VGv8 (6 points)**: The concerns centered on inference latency, benchmark wise performance variance, and missing annotations. Our response included: (1) Supplementary latency measurements across multiple token scales, clarifying when the acceleration benefits are maximized. (2) A detailed temporal duration analysis explaining MARC's expected behavior and its particular strengths on short-to-medium length datasets. (3) A revised manuscript with all corrected figure annotations.

Reviewer **CzmY (6 points)**: This reviewer questioned long video handling and the extent of latency improvements. In response, we used the new experiment (1-frame vs. 8-frames compression on VideoMME) to demonstrate the improvement on long videos with increasing compression budget. Furthermore, the supplementary experiment on latency measurement across multiple token scales could again explain when acceleration is maximized.

Reviewer **CX8T (2 points)**: This reviewer requested further comparisons (VisionZip), a clearer RL motivation, explicit use case descriptions, and clarification on the "1-frame token count". We updated the manuscript accordingly and clarified all conceptual points in detail. Crucially, following our active discussion, the reviewer acknowledged that **"most of my concerns have been resolved"** and explicitly stated an **"inclined to raise the score."**

In conclusion, the feedback from the reviewers has been invaluable, leading to a strengthened paper. We truly appreciate the time and effort invested in the discussion, especially the engagement from Reviewer CX8T. We believe our revised manuscript and comprehensive responses have successfully clarified our contributions and mitigated the noted weaknesses.

Best regards,

The Authors

---

### Public Comment · ~Peiran_Wu2 · 2026-06-08
**Link of code**

https://github.com/Memories-ai-labs/MARC

---

### Meta-Review · Area_Chair_Ld9W · 2026-01-08

**Summary:**

This paper received 4 reviews. The reviewers (score/confidence) are: n2Ge (8/4), CX8T (2/5), VGv8 (6/4), CzmY (6/4). Their major concerns:
- Methodology:
  - Lack of comparisons with key related works (e.g., visionzip, videoxl, video-r), making results less persuasive; weak motivation for RL-based distillation (unclear why RL is chosen over trainable methods) and uncertain conclusion on information loss of training-free methods ``CX8T (2/5)``.
- Experiments:
  - Contradiction in experimental setup description (1 fps sampling with max 64 frames vs. 1 frame in evaluation results) ``CX8T (2/5)``.

**Reviewer Concerns:**

The only negative reviewer, CX8T (2/5), mentioned his/her main concerns are addressed by the rebuttal.

**Reviewer Scores:**

The reviewers (score/confidence) are: n2Ge (8/4), CX8T (2/5), VGv8 (6/4), CzmY (6/4).

After the rebuttal, the only negative reviewer CX8T (2/5) acknowledged that "most of my concerns have been resolved" and explicitly stated an "inclined to raise the score."

---

### Decision · Program_Chairs · 2026-01-26

Accept (Poster)